# Configuration Planning of Expressway Self-Consistent Energy System Based on Multi-Objective Chance-Constrained Programming

Xian Huang, Wentong Ji *, Xiaorong Ye and Zhangjie Feng

School of Control and Computer Engineering, North China Electric Power University, Beijing 102206, China; hx@ncepu.edu.cn (X.H.)
* Correspondence: 120212227007@ncepu.edu.cn

**Abstract:** Regarding the problem of the optimal configuration of self-consistent energy systems based on a 100% renewable energy supply for expressway electricity demand in no-grid areas, this paper proposes a multi-objective planning model based on chance-constrained programming (CCP) to achieve the optimization objectives of low cost and high reliability. Firstly, the number of units of different types of wind turbines (WT), the capacity of photovoltaic (PV) cells, and the number of sets of energy storage systems (ESS) are selected for the design variables in our configuration plan. After defining the load grading shedding and ESS scheduling strategy, the Monte Carlo Simulation (MCS) method and the backward reduction method are applied to model the uncertainties of electric load and renewable energy sources. Finally, the set of Pareto solutions are optimized by the non-dominated sorted genetic algorithm-II (NSGA-II) and its unique best solution is determined by the Criteria Importance Though Intercriteria Correlation (CRITIC) and the Technique for Order Preference by Similarity to Ideal Solution (TOPSIS) approach. Making use of the wind speed and solar radiation intensity historical data of an area in northwest China in the last five years, eight case studies of two typical scenarios are designed and carried out to explore in-depth the impact of different confidence levels and load fluctuation ranges on the planning results. The results verify that the proposed method can effectively improve the robustness of the system and satisfy the power demand in confidence scenarios.

**Keywords:** expressway self-consistent energy system; chance-constrained programming; configuration optimization; NSGA-II; CRITIC; TOPSIS

## 1. Introduction

### 1.1. Background

In the context of the global energy crisis and climate deterioration [1], transportation systems are an important sector of fossil energy consumption. The electrification of transportation has become an important channel to achieve sustainable development, and dependence on the electric power system is gradually increasing [2]. With the progress of high-density battery technology and the increase in charging facilities, the number of electric vehicles has increased rapidly. By March 2022, the number of pure electric vehicles in China reached 7.245 million, a year-on-year increase of 138.20% [3]. However, in some remote or isolated areas without grid access, the spatial layout mismatch between the expressway road network and the grid makes its electrification development difficult.

At the same time, the Chinese transportation system itself contains rich natural endowments [4]. For example, the natural resource endowment of solar energy along China's expressway is $1.023 \times 1012$ kW·h. If these natural resources can be fully utilized, the self-consistency level of the expressway system will be significantly improved. Therefore, the development and popularization of expressway self-consistent energy systems that rely on 100% renewable energy for power generation have become an inevitable trend.

Expressway self-consistent energy is mainly composed of distributed power supplies and energy storage equipment; however, the distributed power sources, mainly WT and PV, are vulnerable to weather factors such as solar radiation, wind speed, etc., which have randomness. Moreover, unlike traditional residential and industrial electrical loads, expressway electrical loads are characterized by shock, volatility, significant temporal characteristics, and high safety requirements. The uncertainty of renewable energy and load makes renewable power abandonment and power shortage frequent [5–7]. Since the flexibility alternatives in the operation stage can be limited, it is necessary to fully consider the above uncertainties in the planning stage and to strive to improve the economy and robustness of the planning scheme under the premise of meeting the demand of the traffic side and the safe and reliable operation of the system, so as to make it suitable for more complex actual operation conditions.

### 1.2. Literature Review

At present, although there are few studies on uncertain optimization methods for expressway self-consistent energy systems, scholars have investigated how to optimize the design of systems by considering uncertainties. The main solutions include stochastic planning and robust optimization.

The basic idea of robust optimization is to use the bounded set model to describe the fluctuation range of uncertain parameters and formulate the optimal decision scheme under the worst scenario according to the set boundary information [8]. Robust optimization is usually used to solve electric vehicle charging station planning (EVCS) problems [9], microgrid optimization dispatch problems [10,11], generation and transmission expansion planning problems [12,13], and power trading with electricity markets problems [14,15]. One of the key factors which affects the difficulty and accuracy of robust optimization solutions is the establishment of uncertainty sets. The existing uncertainty sets are represented by the Box Uncertainty Set [16–18], Polyhedral Uncertainty Set [19], and Ellipsoidal Uncertainty Set [20]. References [21,22] adopt KL divergence and the Wasserstein measure to construct the fuzzy set of uncertain variables. On this basis, the uncertain parameters are taken as optimization variables and solved based on the min-max-min multilayer optimization theory of deterministic optimization. However, in order to avoid the interference of uncertain parameters on the model, the solution of robust optimization is often obtained under the most conservative scenario.

Compared with robust optimization, stochastic planning uses the probability distribution of uncertain variables to model uncertain variables [23] and reduces the conservatism of decision-making. References [24,25] adopt different scenario generation methods to generate typical scenery and establish a two-stage stochastic optimization model for the operation and scheduling of the energy storage system of the hybrid renewable energy system. Reference [26] considers the increasing uncertainty caused by the widespread use of electric vehicles and uses the Monte Carlo simulation and Kantorovich method to deal with the related uncertainties.

However, general stochastic programming is mainly used to deal with optimization problems where random variables only exist in the objective function, and CCP can change the hard constraints into probabilistic formal constraints to realize the consideration of large probability events of random variables. Hence, it can reduce the impact of low-probability extreme events on the optimal solution and improve the rationality of the optimal solution to a certain extent. CCP is widely used in the optimal scheduling of power systems containing renewable energy. Reference [27] applies chance-constrained programming to the day-ahead scheduling of a multi-microgrid system in an uncertain environment. Reference [28] establishes a novel bi-level optimal dispatching model for the CIES with an EVCS in multi-stakeholder scenarios, which optimizes electric vehicles' charging and discharging behavior. In [29], a multi-objective stochastic planning model based on chance constraints of the energy network is developed to minimize the investment cost and the energy pipeline risk. In [30], a unified opportunity-constrained optimization

framework for island microgrid capacity is proposed and a leader-follower structure is proposed to solve the optimal capacity problem.

Although there are many pieces of research on the operation scheduling and demand-side response of microgrids under uncertain conditions, there are few pieces of research that have been conducted on planning issues considering supply reliability and the hierarchical control of different levels of loads. Meanwhile, in the above-mentioned literature, the simulation of uncertain scenes is relatively crude, generating only a set of probability distribution functions [26] or using Markov Chain Monte Carlo (MCMC) simulation [31] to generate data for one year, failing to consider the influence of climate seasonal distribution on uncertain variables.

Since the result of multi-objective optimization is a series of Pareto solutions, decision-makers are still required to choose the best solution from the Pareto set. Therefore, some studies adopt multi-attribute decision-making (MADM) technologies to sort these Pareto solutions and select the best compromise solution. To determine the optimal capacity of the hybrid energy storage system, Reference [32] uses NSGA-II to obtain the Pareto set and apply the improved TOPSIS to select the optimal solution from the Pareto set. In [33], an integrated fuzzy-AHP/TOPSIS/EDAS/MOORA decision-making model for a 100% renewable energy system is proposed, which considers five indicators: cost, reliability, emissions, and social and terrain standards. Reference [34] proposes an optimal two-stage decision-making procedure for the site selection of wind-photovoltaic-shared energy storage projects using veto identification coupled with the fuzzy MCDM method. Reference [35] investigates hybrid renewable energy systems to find the techno-economic and environmental trade-off solutions with the usage of HOMER software, and then uses the TOPSIS method combined with weighting methods to choose the final design among the Pareto solutions set.

### 1.3. Contributions and Paper Organization

In this paper, in order to maintain the feasibility of decision-making at a certain confidence level while minimizing costs and maximizing power supply reliability, CCP, which can ensure the economy and feasibility of planning results, is used to design the self-consistent energy system. The main contributions of this article are as follows: (1) To be as close as possible to the actual scenario, the scenarios in a year are divided into 12 groups by month for simulation, respectively, and the wind speed and solar radiation are assumed to obey the Weibull distribution and Beta distribution, respectively [36]. (2) Typical scenarios of 8760 h of a year are generated by using MCS and the backward reduction method. (3) Based on the characteristics of expressway load classification, with the operation control strategy combining load grading shedding and the ESS schedule, a multi-objective optimization model is established with the annual cost of the whole life cycle as the economic index and the power supply reliability as the reliability index. (4) The uncertainty model is converted to a scenario-based deterministic model by using CCP theory, and is solved through NSGA-II and CRITIC-TOPSIS.

The rest of the paper is organized as follows. The mathematical models of expressway self-consistent energy system components are presented in Section 2. The method for calculating the probability density of wind speed and solar radiation intensity distribution and generating random scenes is presented in Section 3. Section 4 presents the proposed planning methodology and the algorithm of the methodology. The simulation results are presented and discussed in Section 5. Conclusions and future works are drawn in Section 6.

### 2. System Architecture and Mathematical Model

The components of the renewable energy-based self-consistent system include WT, PV, and ESS, which are shown in Figure 1. According to the requirements for power supply reliability, the electric load of the expressway is divided into three levels [37], of which the Level I load has the highest requirements for power supply reliability. The electrical load levels of highway power equipment are shown in Table 1.

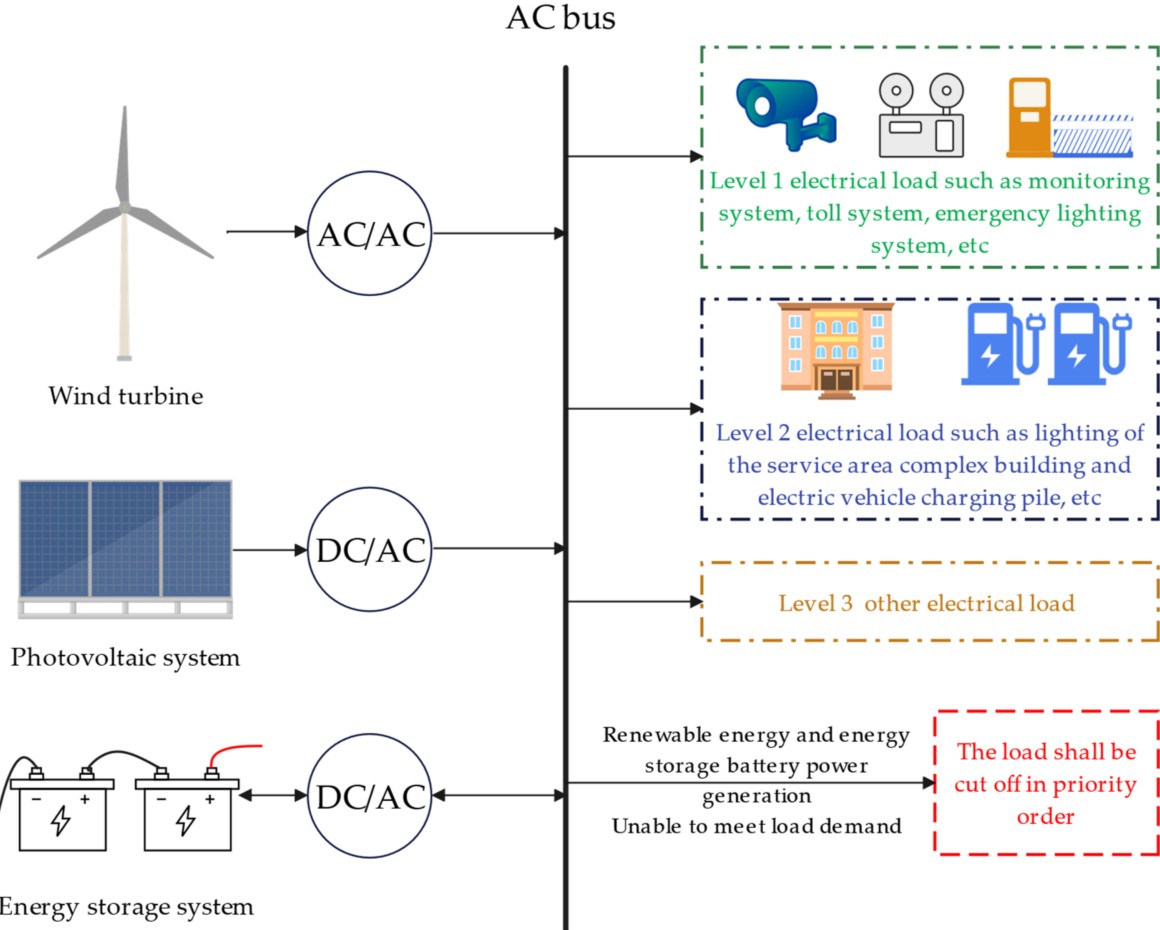

**Figure 1.** Schematic of expressway self-consistent energy system.

**Table 1.** Electricity load level of power-using equipment.

| Electricity Load Level | Electrical Equipment |
| --- | --- |
| Level I | Toll systems; Communication systems; Control room emergency alarm systems for communication systems; Fire protection systems and emergency lighting systems. |
| Level II | The Lighting of the management center and service area; Fire protection systems for general facilities; Electric Vehicle Charging Posts. |
| Level III | Other facilities. |

*2.1. Mathematical Models for the System Components*

2.1.1. Wind Turbine

The wind power generated by a wind turbine at time $t$ can be represented by Equation (1) [38]:

$$P_{WT}(t) = \begin{cases} 0, & v(t) < v_{in}, v(t) > v_{out} \\ P_r \frac{v(t)-v_{in}}{v_r-v_{in}}, & v_{in} \leq v(t) \leq v_r \\ P_r, & v_r \leq v(t) \leq v_{out} \end{cases} \tag{1}$$

where $P_r$ is the rated power; $v(t)$ is the actual wind speed at time $t$ at the turbine hub; $v_{in}$, $v_r$, and $v_{out}$ are the cut-in, rated, and cut-out wind speed, respectively.

In general, the known wind speed data are at the height of the wind measurement tower; therefore, the data must be converted to the actual wind speed at the turbine hub using the following equation, Equation (2):

$$v(t) = v_{std}(t) \times \left( \frac{H_{hub}}{H_{std}} \right)^{\theta} \tag{2}$$

where $v_{std}(t)$ is the wind speed at time $t$ at the wind measurement tower; $H_{hub}$ and $H_{std}$ are the height of the turbine hub and wind measurement tower; $\theta$ is the friction coefficient, which is taken as 0.2 in this paper.

### 2.1.2. Photovoltaic System

The output power of photovoltaic panels can be calculated by Equation (3) [38]:

$$P_{PV}(t) = f^{PV} \times C_{PV} \times \frac{I(t)}{I^{STC}} \left[ 1 + \gamma \left( T^{PV}(t) - T^{PV-R} \right) \right] \tag{3}$$

where $C_{PV}$ is the rated power of the PV panel under standard test condition (STC); $I(t)$ is the actual solar radiation intensity on the PV panel at time $t$ and $I^{STC}$ is the solar radiation intensity at the STC; $f^{PV}$ is the PV derating factor due to the changing effect of the temperature and dust on the panels; $\gamma$ is the temperature coefficient of power; $T^{PV}(t)$ and $T^{PV-R}$ are the real-time and STC of the PV panel temperatures, respectively.

### 2.1.3. Energy Storage Battery

The selection of an appropriate size of battery bank requires a complete analysis of the charge/discharge process of the battery. The main parameter of the battery to be considered is the *EC* (equals energy capacity), which is simulated during the charging process as [25]:

$$E(t+1) = E(t) + \left( \phi^{ch} P_{ch}(t) \Delta t - \frac{1}{\phi^{dis}} P_{dis}(t) \Delta t \right) \tag{4}$$

where $P_{ch}(t)$ and $P_{dis}(t)$ represent charging power and discharging power; $\phi^{ch}$ and $\phi^{dis}$ are charging and discharging efficiency, respectively; $\Delta t$ is the simulation step, at 1h.

### 2.2. Power-Flow Strategy

When the energy storage battery is not put into operation, the value of the power imbalance is $\Delta P(t)$ of the system, as shown in Equation (5):

$$\Delta P(t) = P_{WT}(t) + P_{PV}(t) - (L_1(t) + L_2(t) + L_3(t)) \tag{5}$$

where $L_1(t)$, $L_2(t)$, and $L_3(t)$ are the power of level I, level II, and level III of the electrical load at moment $t$, respectively.

The generation power of the energy self-supply system can be simulated with Equation (6):

$$P_{sys}(t) = \begin{cases} P_{WT}(t) + P_{PV}(t), (a) \\ \\ P_{WT}(t) + P_{PV}(t) + P_{dis}(t), (b) \end{cases} \tag{6}$$

(a)  If $\Delta P(t) \geq 0$, the total power generated by wind turbine and PV is sufficient to cover the load demand.

(b)  Otherwise, when $P_{WT}(t)$ and $P_{PV}(t)$ are not sufficient to meet the demand, the battery supplies the difference. If the energy storage battery cannot meet the load demand, the load will cut off according to the order of Level III, Level II, and Level I. In this paper, $Ls_1(t)$, $Ls_2(t)$, and $Ls_3(t)$ are the load shedding of 3 levels of electrical load, respectively. The situation is classified as follows:

Case 1: $P_{sys}(t) \geq L_1(t) + L_2(t) + L_3(t)$, there is no power shortage.

Case 2: $P_{sys}(t) < L_1(t) + L_2(t) + L_3(t) \cap P_{sys}(t) \geq L_1(t) + L_2(t)$, there are power shortages in the Level III load, with the missing quantity determined based on Equation (7).

$$Ls_3(t) = -(P_{sys}(t) - L_1(t) - L_2(t) - L_3(t)) \times \Delta t \tag{7}$$

Case 3: $P_{sys}(t) < L_1(t) + L_2(t) \cap P_{sys}(t) \geq L_1(t)$, there are power shortages in the level III and Level II loads, with the missing quantity determined based on Equation (8).

$$\begin{cases} Ls_3(t) = L_3(t) \times \Delta t \\ \\ Ls_2(t) = -(P_{sys}(t) - L_1(t) - L_2(t)) \times \Delta t \end{cases} \tag{8}$$

Case 4: $P_{sys}(t) < L_1(t)$, there is power shortage in the Level I to Level III electric load, with the missing quantity determined based on Equation (9).

$$\begin{cases} Ls_3(t) = L_3(t) \times \Delta t \\ \\ Ls_2(t) = L_2(t) \times \Delta t \\ \\ Ls_1(t) = -(P_{sys}(t) - L_1(t)) \times \Delta t \end{cases} \tag{9}$$

The specific operating strategy of the expressway self-consistent energy system is shown in Figure 2.

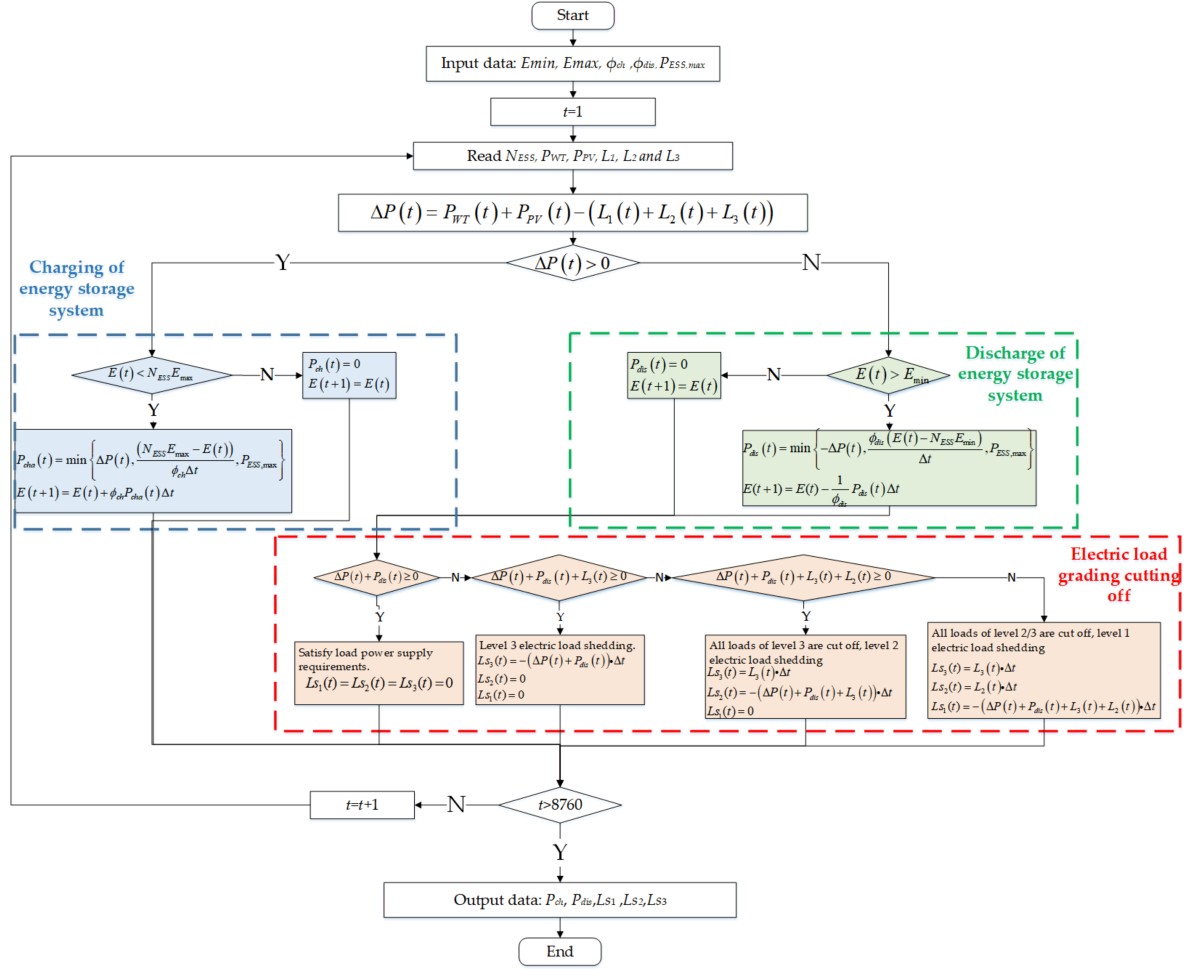

**Figure 2.** Operating strategy of expressway self-consistent energy system.

## 3. Uncertainty Consideration

### 3.1. Preprocessing Uncertain Data

To represent the actual situation, a year is divided into 12 parts by months and each month is made to be 30 days long for convenience. Therefore, this paper divides the historical data by month and fits the wind speed and solar radiation intensity data into Weibull distribution and Beta distribution, respectively, to obtain the probability density function (PDF) under each time slice. In the proposed model, each uncertainty variable has 12 24 probability distributions. Figure 3 shows the PDF of 24 h of a day in January.

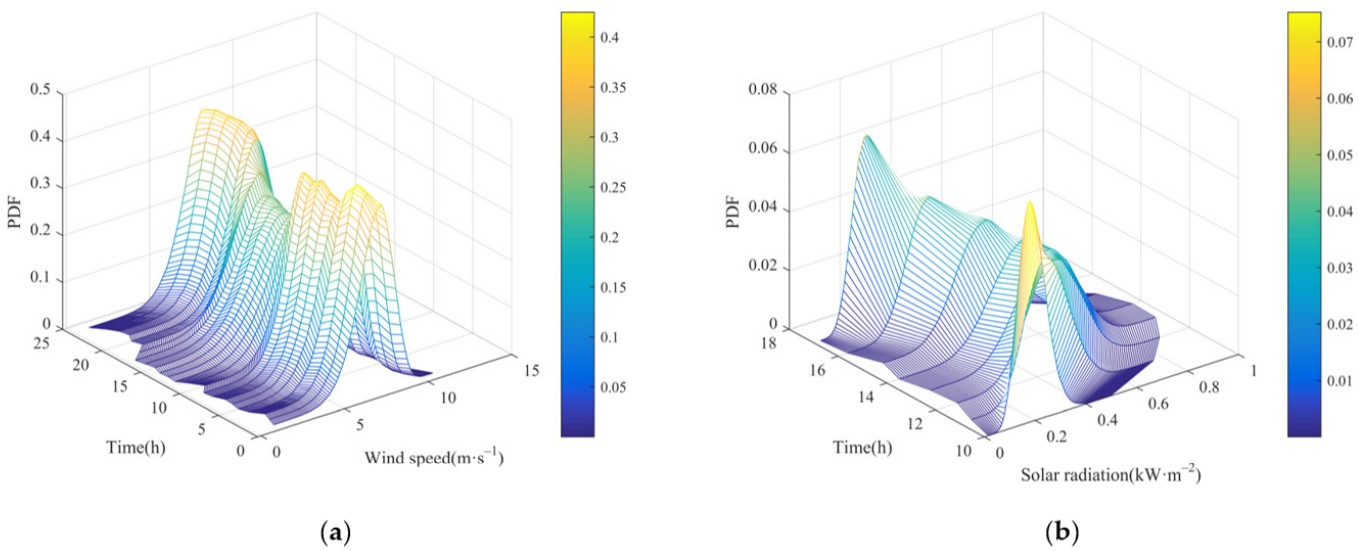

(**a**)  (**b**)

**Figure 3.** The Probability density function of the renewable resource. (**a**) PDF of wind speed; (**b**) PDF of solar radiation.

### 3.2. Scenario-Generation Methods

Based on the obtained PDF, the MCS is used to generate a large number of scenarios. To ensure the processability of the calculation in the planning process, it is necessary to use the backward method to reduce the scenarios and retain the most representative typical scenarios in the sample cluster, as well as a few more extreme scenarios to solve the planning problem in a limited scenario. The implementation of the employed scenario reduction method for *N* scenarios can be explained as follows [39,40]:

Step 1: Determine $S$ as the scenarios set. Set $\pi_s$ as the probability of each scenario, and $Dt(S_m, S_n)$ as the distance of $(S_m, S_n)$.

Step 2: Calculate the distance between every two scenarios:

$$Dt(S_m, S_n) = \sqrt{\sum_{i=1}^{d} \left(x_i^{s_m} - x_i^{s_n}\right)^2} \tag{10}$$

where $d$ is the length of the time series of each scenario.

Step 3: Calculate the minimum distance between $S_r$ and $S_k$, based on the following equation.

$$Dt_{k,r} = \min_{i=1,2,..,N \cap i \neq k} Dt(S_k, S_i) \tag{11}$$

Step 4: In this step, $Dt_{k,r}$ and $PD_{k,r}$ are determined based on Equations (12) and (13).

$$PD_{k,r} = \pi_k Dt_{k,r} \tag{12}$$

$$PD_j = \min PD_k \tag{13}$$

Step 5: Delete scenario $S_j$ and update the probability of each scenario after reduction:

$$S = S - \{S_j\}, \pi_r = \pi_r + \pi_j \tag{14}$$

Step 6: Repeat steps 2 to 4 until the ideal number of scenarios is obtained.

## 4. Problem Formulation and Design Methodology

### 4.1. Objective Function and Constraints

In the proposed optimization process, the equivalent annual cost of the life cycle of the system and the reliability of the power supply (RS) have been accounted for as objective functions.

### 4.1.1. Equivalent Annual Cost in Life Cycle

The objective function of the proposed mathematical model involves minimizing the total cost, which consists of investment, and operating and maintenance (O&M) costs. To simplify the calculation, this paper assumes that the load, wind speed, and solar radiation levels remain consistent from year to year during the project planning period, and simplifies the total cost of the whole life cycle to the cost of one year, which is formulated as Equation (15):

$$\min TC = CRF(i, R_{pro})C_{IC} + C_{O\&M} \tag{15}$$

$$CRF(i, R_{pro}) = \frac{i(1+i)^{R_{pro}}}{(1+i)^{R_{pro}} - 1} \tag{16}$$

where $TC$ is the total annualized cost (¥); $C_{IC}$ and $C_{O\&M}$ are the investment operation and maintenance costs (¥), respectively; $i$ is the interest rate, at 0.1; $R_{pro}$ is the project lifetime, at 20 years. The capital recovery factor is presented by $CRF(i, R_{pro})$.

The specific investment costs and O&M costs can be calculated as follows:

$$C_{IC} = c_{WT}^{inv}N_{WT} + c_{PV}^{inv}C_{PV} + c_{ESS}^{inv}N_{ESS} \tag{17}$$

$$C_{O\&M} = \sum_{t=1}^{8760} \left( c_{WT}^{pro}P_{WT}(t)\Delta t + c_{PV}^{pro}P_{PV}(t)\Delta t + c_{ESS}^{pro}(P_{ch}(t) + P_{dis}(t))\Delta t \right) \tag{18}$$

where $c^{inv}$ is the unit investment cost factor; $c^{pro}$ is the operation and maintenance cost factor.

### 4.1.2. Reliability of Power Supply

When planning an independent renewable energy power system based on the power demand of an expressway, there are specific requirements for power supply reliability. Moreover, power supply reliability is closely related to the uncertainty of renewable energy generation. Therefore, this paper takes the power supply reliability rate as the optimization objective, which is determined as follows:

$$\max RS = \left( 1 - \frac{T_{user}}{T_{st}} \right) \times 100\% \tag{19}$$

where $T_{user}$ is the average power failure time; $T_{st}$ is the statistical period time, and equals 8760 h.

### 4.1.3. Area Constraints

Due to the limited available area around the expressway and the influence of terrain, it is necessary to limit the maximum number of allowable configurations of various distributed power sources.

$$\begin{cases} 0 \leq N_{WT1} \leq N_{WT1,max} \\ \\ 0 \leq N_{WT2} \leq N_{WT2,max} \\ \\ 0 \leq C_{PV} \leq C_{PV,max} \\ \\ 0 \leq N_{ess} \leq N_{ess,max} \end{cases} \tag{20}$$

#### 4.1.4. ESS Operation Constraints

The following denotes the formula for charging and discharging power, in which the charged energy cannot exceed the maximum energy capacity of ESS.

$$P_{ch}(t) = max\left\{ min\left\{ \Delta P(t), \frac{N_{ESS}E_{max} - E(t)}{\phi_{ch}\Delta t} \right\}, N_{ESS}P_{ESS,max} \right\} \tag{21}$$

$$P_{dis}(t) = max\left\{ min\left\{ -\Delta P(t), \frac{\phi_{dis}(E(t) - N_{ESS}E_{min})}{\Delta t} \right\}, N_{ESS}P_{ESS,max} \right\} \tag{22}$$

$$N_{ESS}E_{min} \leq E(t) \leq N_{ESS}E_{max} \tag{23}$$

where $E_{max}$ and $E_{min}$ are the maximum and minimum energy capacity of a single ESS, respectively; $P_{ESS,max}$ is the maximum charging and discharging power of single energy storage equipment.

#### 4.1.5. Loss of Power Supply Probability Constraints (LPSP)

In the process of expressway use, the interruption of power supply to Level I and II loads will cause great economic losses, affect the normal work of important units, or even cause personal injury. Thus, this paper makes the following constraints on the shedding amount of Level I and II loads [41].

$$\begin{cases} \text{LPSP}_1 = \frac{\sum_{t=0}^{T} Ls_1(t)}{\sum_{t=0}^{T} L_1(t)} \leq 1\% \\ \\ \text{LPSP}_2 = \frac{\sum_{t=0}^{T} Ls_2(t)}{\sum_{t=0}^{T} L_2(t)} \leq 5\% \end{cases} \tag{24}$$

### 4.2. Chance-Constrained Programming

CCP is mainly used to deal with mathematical problems where the constraints or objective functions contain random variables. Since, in some unfavorable scenarios, the decisions made may not satisfy the objective function or constraints, a confidence level is set for the planning, allowing the decisions to not satisfy the constraints to a certain extent; however, the probability of satisfying the constraints must be guaranteed not to be lower than this confidence level [42]. The general form of multi-objective chance-constrained programming is [43]:

$$\begin{cases} min[\overline{f_1}, \overline{f_2}, ..., \overline{f_m}] \\ \\ s.t. \\ \\ Pr\left\{ f_i(x, \xi) \leq \overline{f}_i \right\} \geq \alpha, i = 1, 2, 3... \\ \\ Pr\left\{ g_j(x, \xi) \leq \overline{g}_j \right\} \geq \beta, j = 1, 2, 3... \end{cases} \tag{25}$$

where $f$ is the objective function; $g$ represents the inequality constraints; $x$ and $\xi$ are the vectors of state and uncertain variables; $\alpha$ and $\beta$ are pre-defined confidence levels for the objective function and constraints.

By using CCP, the optimization problem in this paper can be defined as:

$$\begin{cases} min[\overline{TC}, \overline{-RS}] \\[6pt] s.t. \\[6pt] Pr\{TC_s \leq \overline{TC}\} \geq \alpha \\[6pt] Pr\{RS_s \geq \overline{RS}\} \geq \alpha \end{cases} \tag{26}$$

where $TC_s$ and $RS_s$ are the values of the objective function under scenario $s$; $x$ and $\xi$ are the vectors of state and uncertain variables; $\alpha$ and $\beta$ are pre-defined confidence levels for the objective function and constraints.

The probabilistic constraint in Equation (24) can be changed to a deterministic constraint as follows:

$$\begin{cases} Pr(\text{LPSP}_{1,s} \leq 1\%) \geq \alpha \\[6pt] Pr(\text{LPSP}_{2,s} \leq 5\%) \geq \alpha \end{cases} \tag{27}$$

where $\text{LPSP}_{1,s}$ and $\text{LPSP}_{2,s}$ are the values of the constraints under scenario $s$.

The calculation of the objective function and constraints at specific confidence levels is shown below:

Step 1: $N$ random scenarios are generated and substituted into the model, and the objective function values and constraint values are calculated for each scenario.

Step 2: According to the law of large numbers, if the minimum value of the objective function is required, the values under the $N$ scenarios calculated by Step1 are sorted from smallest to largest, and the value of the $M_{\text{th}}$ element is used to estimate the minimum value $f$ of the confidence level $\alpha$ of the objective function; otherwise, the values of the objective function under each scenario must be sorted from largest to smallest.

Step 3: According to the law of large numbers, the probability measure of constraint satisfaction $Pr$ is defined as the sum of the probabilities of occurrence of scenarios that satisfy the constraints. The $Pr$ value of the constraint $g(x,\xi)$ can be calculated by the following equations.

$$h(\xi_j) = \begin{cases} \pi_j, & \text{if } g(\xi_j) \leq 0 \\ 0, & \text{otherwise} \end{cases} \tag{28}$$

$$Pr\{g(\xi) \leq 0\} = \sum_{j=1}^{N} h(\xi_j) \tag{29}$$

### 4.3. Implementation of the Proposed Algorithm

The methodology of finding the optimum size of a standalone WT/PV/battery bank by following the chance constrained approach is discussed in this section. The procedures of the proposed algorithm can be summarized in the following steps:

1.  All the required data of the problem are inputted, which include historical data on wind speed, solar radiation intensity and load, microgrid equipment (distributed power supply and energy storage system) parameters, and market price.
2.  Based on the theory described in Section 3.1. and Section 3.2., the wind speed and solar radiation probability density distribution parameters are obtained, and the MCS technology is used to generate a certain amount of scenarios (here, 2000 scenarios). Then, the backward method is applied to reduce the number of generated scenarios (here, scenarios have been reduced to 20). Figure 4 shows the reduction of the 24-h wind speed scenario, taking one day in January as an example.

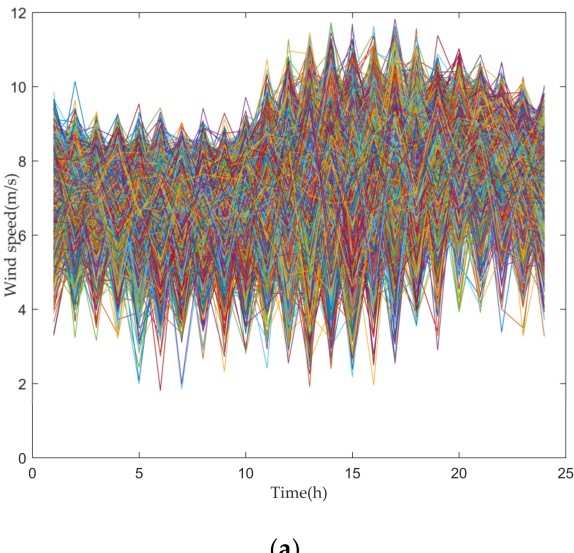
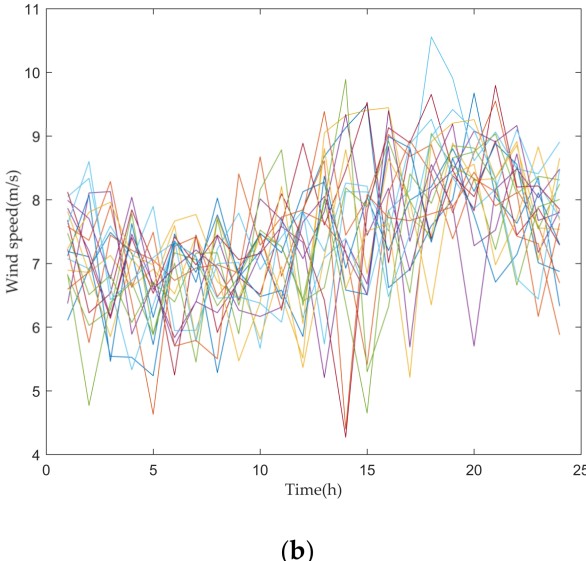

(**a**)
(**b**)

**Figure 4.** Wind speed scene set. The 24-h wind speed scenarios represented by colorful lines. (**a**) Before reduction (there are 2000 scenarios represented by 2000 colorful lines); (**b**) After reduction (there are 20 scenarios represented by 20 colorful lines).

3.  In this paper, NSGA-II is used for multi-objective capacity optimization configuration. The optimization variables of the system are as follows: $C_{PV}$ is the rated power of photovoltaic panels, $N_{WT1}$ and $N_{WT2}$ are the numbers of type I and II wind turbines, and $N_{ESS}$ is the number of batteries. The optimization objectives under the confidence level are shown in Equation (26). The constraints are shown in Equations (20)–(23) and (27).

4.  The weights quantify the importance and priority of each attribute and have a significant impact on the decision results [44]. In this paper, the CRITIC method is utilized to determine the weights of the two objectives and the TOPSIS method is used to rank the Pareto solutions, and is widely applied to solve problems in energy planning [45], system potential assessment [46], robot selection [47], etc.

The flow chart of the above algorithm is shown in Figure 5.

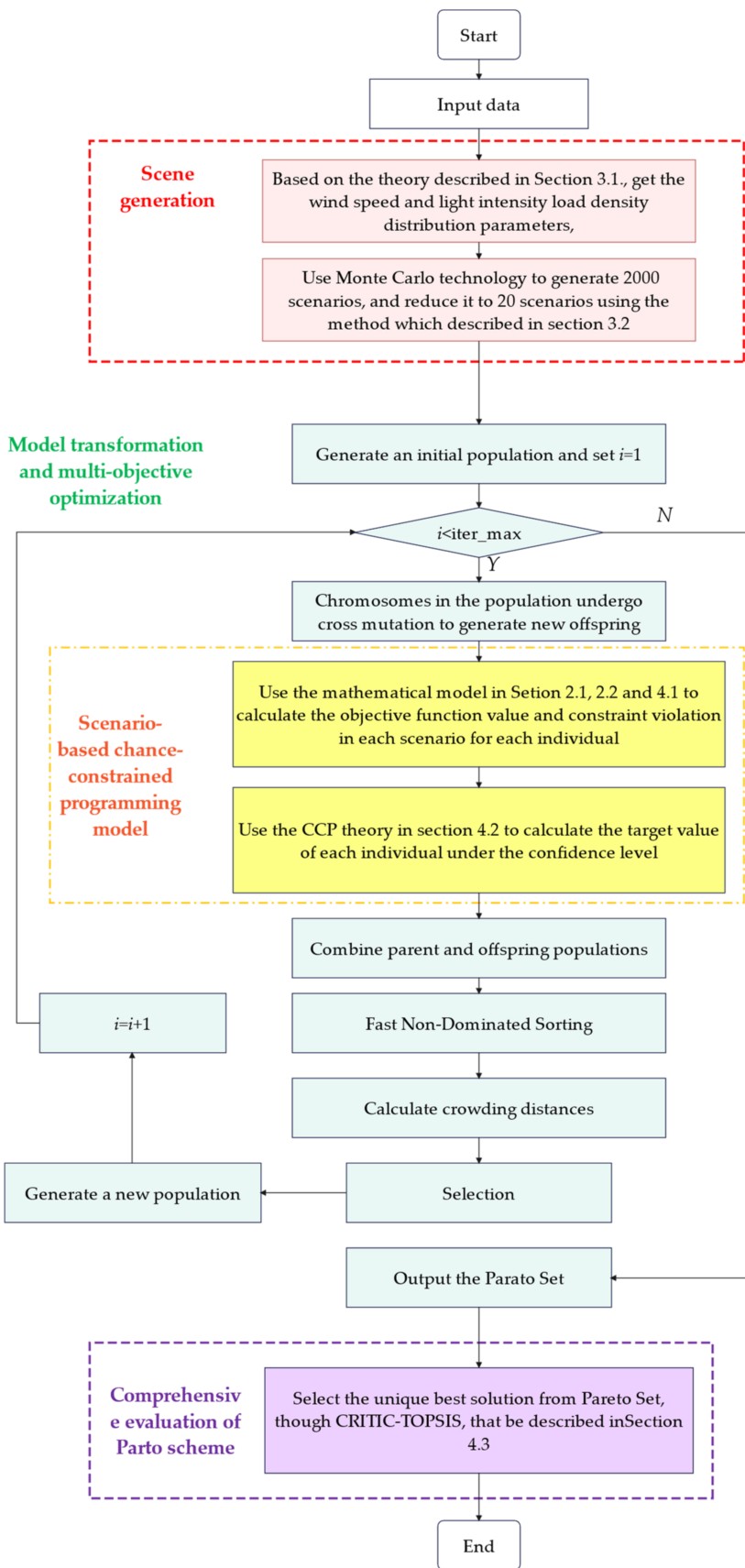

**Figure 5.** Algorithm flow chart.

## 5. Case Study

### 5.1. Data Description

In this paper, the planning model is simulated and analyzed based on the data from 2017 to 2022 for a region in northwest China. The input data include wind speed, solar radiation, and load consumption. Tables 2–4 show the equipment parameters and market prices for the different devices, respectively.

**Table 2.** Wind turbine parameters.

| Equipment Parameters | Type 1 | Type 2 |
| --- | --- | --- |
| Rated power (kW) | 100 | 500 |
| Cut-in wind speed (m/s) | 2.5 | 2.5 |
| Rated wind speed (m/s) | 10 | 10 |
| Cut-out wind speed (m/s) | 45 | 45 |
| Hub height (m) | 10.5 | 18 |
| Investment cost (¥) | 280,000 | 1,450,000 |
| Operation and maintenance cost (¥/kWh) | 0.2 | 0.23 |

**Table 3.** PV panel parameters.

| Equipment Parameters | Numerical Value |
| --- | --- |
| PV de-rating factor | 0.8 |
| Rated capacity (kW) | 1 |
| Electrical conversion efficiency of the PV panel | 0.13 |
| Temperature coefficient of power | −0.005 |
| STC of the PV panel temperature (°C) | 25 |
| Investment cost (¥) | 6195 |
| Operation and maintenance cost (¥/kWh) | 0.1 |

**Table 4.** Battery equipment parameters.

| Equipment Parameters | Numerical Value |
| --- | --- |
| $E_{\max}$ (kWh) | 138.24 |
| $E_{min}$ (kWh) | 0 |
| $P_{ESS,max}$ (kW) | 50 |
| Charge and discharge efficiency | 0.86 |
| Investment cost (¥) | 55,200 |
| Operation and maintenance cost (¥/kWh) | 0.2 |

### 5.2. Optimization Results

The Pareto frontier of NSGA-II under 90% confidence is shown in Figure 6. From Figure 6a, it is easy to see that there is a clear conflict between the two objectives, and there is no feasible solution that can simultaneously reduce the cost and increase the supply reliability. However, as the equivalent annual value cost increases, the growth rate of RS gradually decreases, i.e., diminishing marginal utility. In order to further analyze the impact of different optimal configuration results on the system, three representative Schemes are selected from the Pareto solution set and their economic and reliability indicators are compared and analyzed. Among the three Schemes in the figure, Scheme 1 has the lowest annual system cost and Scheme 3 has the highest supply reliability. Scheme 2 is in between Schemes 1 and 2, and is the optimal compromise obtained by using the CRITIC-TOPSIS optimization method.

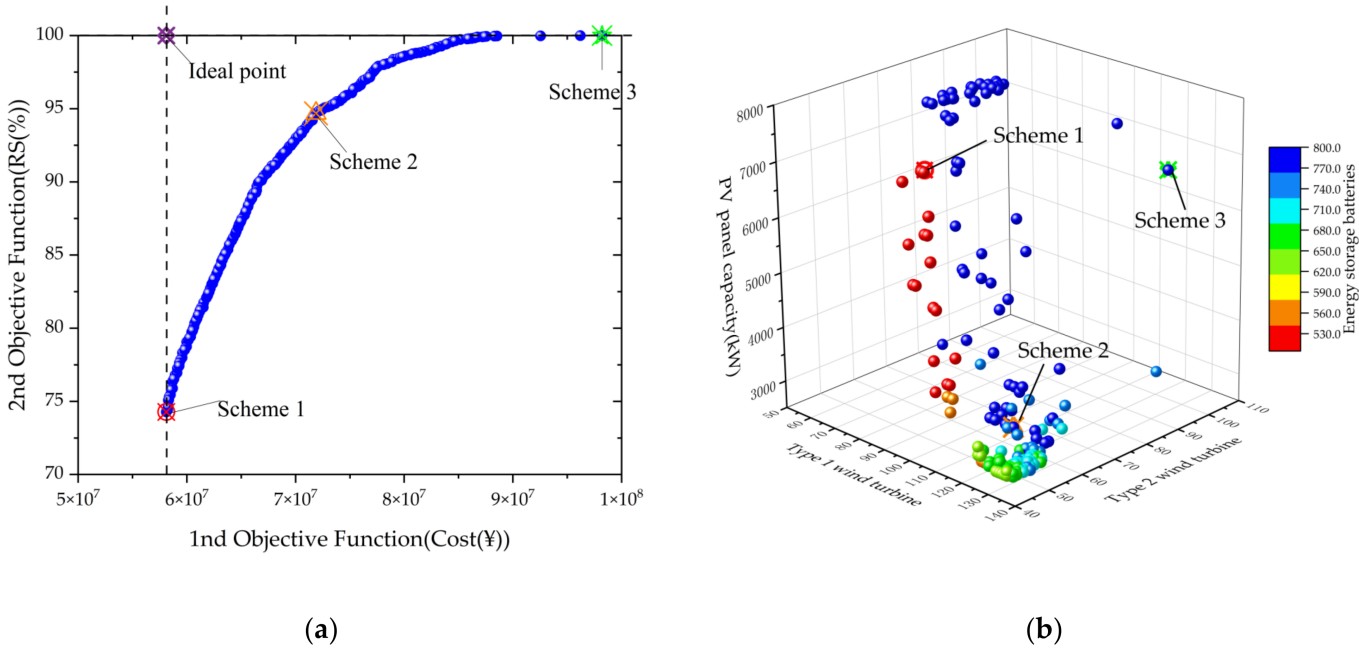

(**a**)                                                                (**b**)

**Figure 6.** Pareto solutions. (**a**) Objective function; (**b**) design variables.

Figure 7 shows the operation process in a day (24h) under the above three design schemes. Under Scheme 1, the system will have Level III, II, or even I load shedding, while under Scheme 2, load shedding rarely occurs. Under Scheme 3, the demand for all loads can be met; however, the waste of renewable energy is serious, that is, the utilization rate of renewable energy is low. Therefore, the CRITIC-TOPSIS method is used to select the best compromise scheme with high power supply reliability and low energy waste to meet the actual demand.

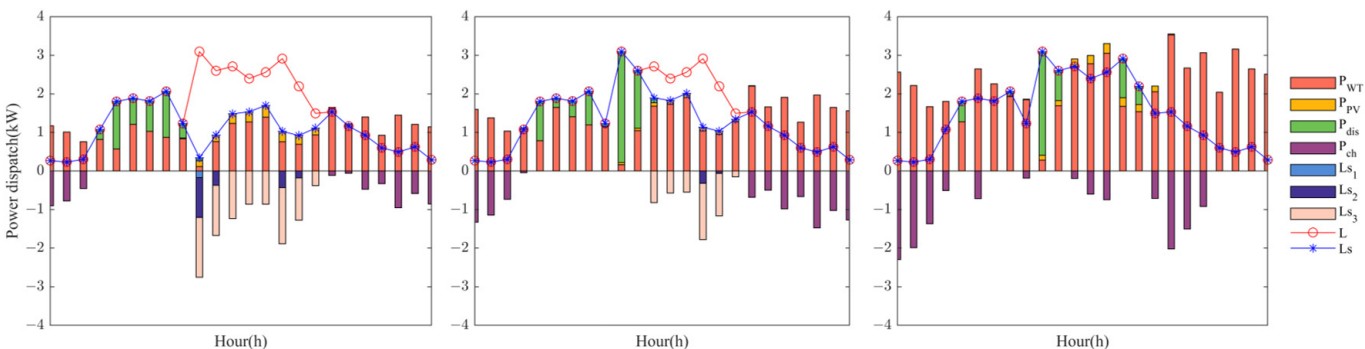

**Figure 7.** Distributed energy output and load shedding results in Scenario I.

Table 5 shows the specific system equipment configuration results and characteristic index values at the confidence level of the three schemes. It can be seen in Table 5 that, from Scheme 1 to Scheme 2, the number of energy storage batteries increases the most, because the increase in the number of energy storage batteries can effectively reduce the power imbalance between the power supply side and the power consumption side, and improve the system operation stability. However, from Scheme 2 to 3, due to the limitation of the installed capacity of the energy storage equipment, it is necessary to reduce load shedding by increasing the installed capacity of the renewable energy.

**Table 5.** System configuration of each scheme and evaluation index.

| Configuration Results and Metrics | | Scheme 1 | Scheme 2 | Scheme 3 |
|---|---|---|---|---|
| Number of Type 1 wind turbine | | 106 | 113 | 114 |
| Number of Type 2 wind turbine | | 41 | 61 | 109 |
| Capacity of PV panel | | 7682 kW | 2797 kW | 6348 kW |
| Number of Energy storage batteries | | 506 | 790 | 766 |
| Equivalent annual cost in life cycle ($\times 10^7$ ¥) | | 5.7114 | 7.1890 | 9.8198 |
| Reliability of power supply (%) | | 74.28% | 94.68% | 100% |
| LPSP/% | Level I | 0.24 | 0.02 | 0 |
| | Level II | 5.00 | 1.12 | 0 |
| | Level III | 26.27 | 4.74 | 0 |

Figure 8 shows the variation in the battery EC over the year for each scheme. As can be seen, although the design schemes are different, the variation in the battery EC in all cases is similar. In these cases, the battery is charged when the power load is less at night, and is discharged rapidly during the day, the peak period of power consumption. Moreover, the value of the EC has obvious seasonal differences. The battery storage system starts to become fully charged in March. Then, the battery has run down in October. This shows that the generated random scenarios have modeled the stochastic behavior of uncertainty sources properly.

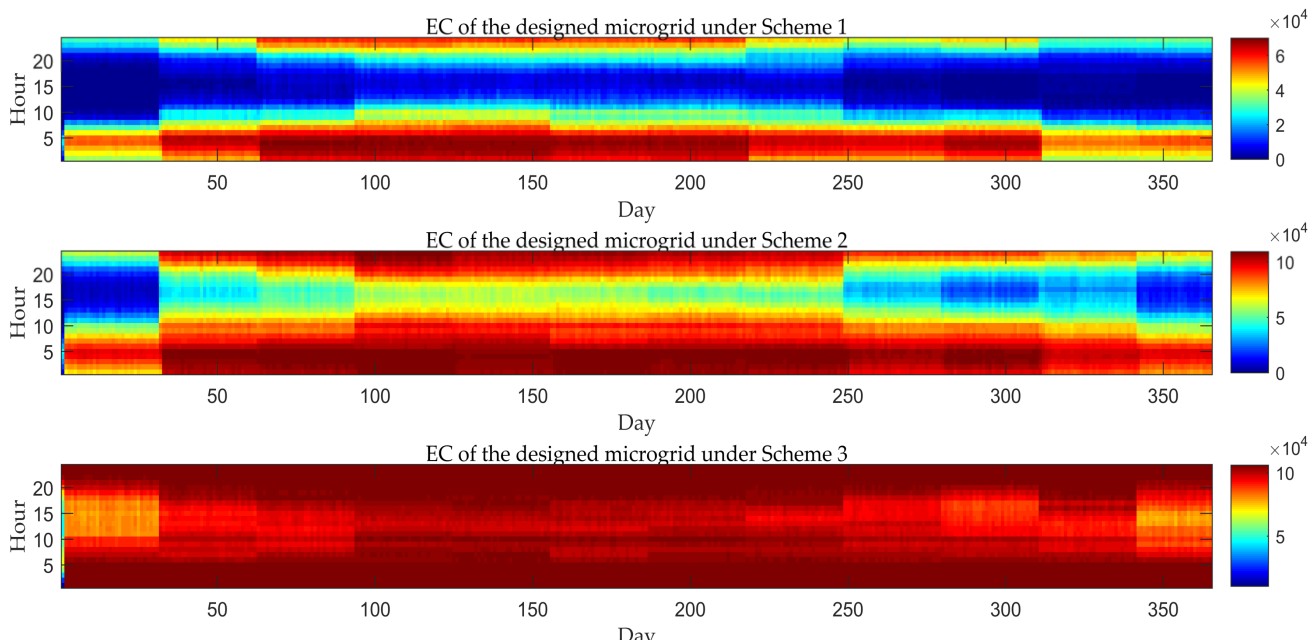

**Figure 8.** The EC of battery changes in a year.

### 5.3. The Effect of the Confidence Levels of the CCP Model

The level of confidence indicates the degree to which the system is systematic in different scenarios: the higher the confidence level, the higher the requirement for system robustness. In this paper, the confidence level $\alpha$ is set as 80%, 85%, 90%, and 95%, respectively. The total cost of optimum solutions of different RS (92%, 94%, 96%, 98%, and 100%) for different confidence levels are shown in Figure 9.

It can be seen from Figure 9, with the increase in $\alpha$, that the total cost of the system will increase. Further, when RS is in the four cases of 92–98%, the growth rate will obviously increase after 90% confidence and change from less than 1% to about 2% or even 3%; when RS is taken as 100%, the growth rate will increase significantly after 85% confidence. Therefore, in order to avoid the high cost caused by the high confidence level of opportunity

constraints, the best confidence level can be selected according to the degree of demand for power supply reliability. In this case, the best confidence level is 85% or 90% under different RS. In practical application, the optimal confidence level will vary depending on the data and operation control strategy.

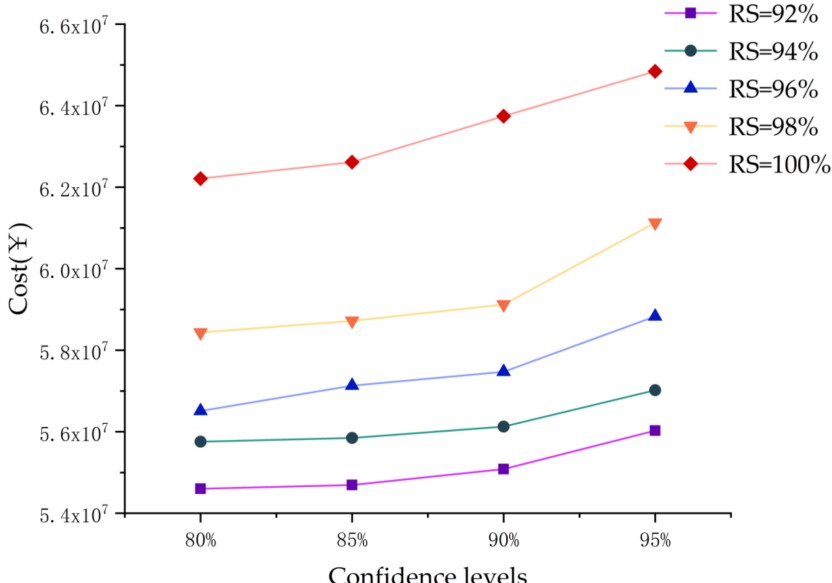

**Figure 9.** The total cost of optimum solutions of different RS for different confidence levels.

### 5.4. The Effect of the Load Characteristics on Planning Results

In this paper, the load is set to follow the normal distribution, and its distribution is specifically expressed as $P_l \sim N(\mu, k \cdot \mu)$, where $k$ is the coefficient of variance. The larger $k$ is, the larger the load fluctuation range is. In this paper, the $k$ values are set as 0.1, 0.3, 0.5, and 0.7, respectively.

Figure 10 shows the design variable values of the Pareto solution set under different load fluctuation ranges. It can be seen, when the setting of $k$ is different, that the planning has obvious differences. On the whole, the larger the $k$ is, the larger the load fluctuation range is, the higher the robustness requirements of the scheme are, and the higher the number of distributed power generation and energy storage batteries will be.

As can be seen from Figure 10d, the number of energy storage batteries quickly reaches saturation. This is because only the storage batteries in this system can coordinate the imbalance between the source and the load, which has a great impact on the reliability of the system's power supply; therefore, the capacity of the storage batteries gradually tends to the maximum installed capacity. In the distributed power supply, type II WT has the largest installed capacity, followed by type I WT, and, finally, PV. Further, with the increase in load fluctuation range, the number of type II wind turbines with a rated power of 500kW increases significantly, while the number of other distributed power sources has not changed much or even slightly decreased. Therefore, compared with other distributed power sources, type II WT are more suitable for the application scenarios in this paper, and when the power demands on expressway increase, priority should be given to increasing the number of type II WT.

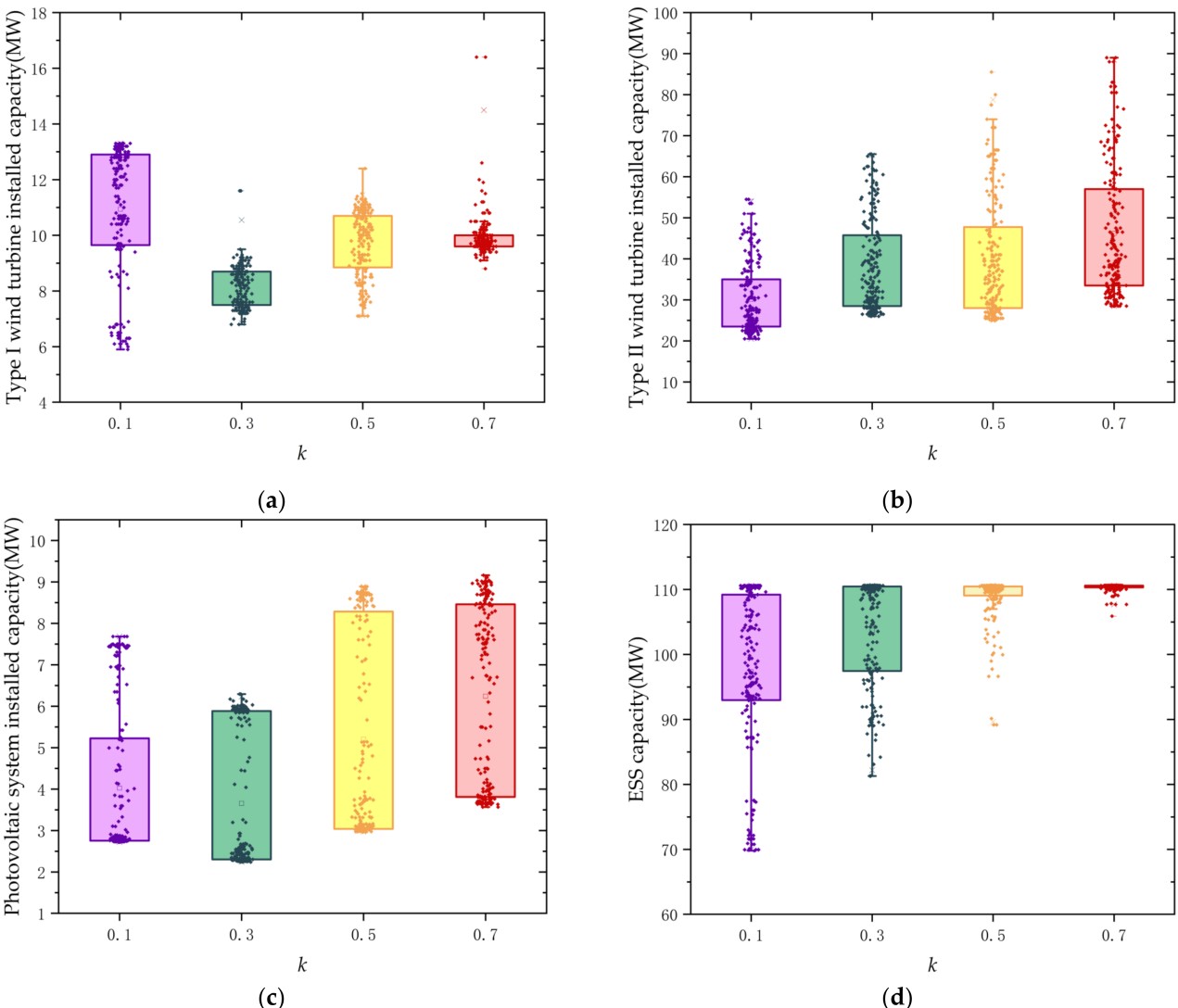

**Figure 10.** Pareto value of design variable under different k values which are represented by different colors. (**a**) Type I wind turbine installed capacity; (**b**) type II wind turbine installed capacity; (**c**) PV installed capacity; (**d**) ESS installed capacity.

## 6. Conclusions and Future Works

In this paper, the multi-objective capacity optimization problem of expressway self-consistent energy systems based on the uncertainty of wind speed, solar radiation, and electrical load is studied. With the use of CCP optimization theory, the best solution which considers both power supply reliability and economy in Pareto solutions is determined by NSGA-II and CRITIC-TOPSIS. Moreover, the optimization model is an investment decision and equipment selection problem, which has good engineering applicability. However, this method must be based on the accurate PDF of uncertain parameters such as wind speed and solar radiation, as well as electrical load demand. Also, constructing the PDF requires obtaining a large amount of sample information. For many practical problems, it may be difficult to obtain sufficient uncertainty information, leading to deviations in planning results.

In future work, we will carry on further research on the following points:

- The practical engineering application is definitely more complex than the case studies in this paper, entailing more design variables and constraints as well as a larger system capacity that even requires a group of microgrids to achieve energy self-consistency.

- The CRITIC method is highly dependent on sample data and cannot reflect the importance that decision-makers attach to different attribute indicators. In application, it is meaningful to obtain a more scientific weight value by utilizing expert experience and professional knowledge.
- For practical application, it is necessary to obtain the specific and detailed resource data of wind speed and solar radiation as well as electrical load demand of the planning area, as the optimized planning results are highly dependent on these data.

**Author Contributions:** Conceptualization, X.H.; methodology, X.H. and W.J.; software, X.H. and W.J.; validation, X.H., X.Y. and W.J.; data curation, X.Y. and Z.F.; writing—original draft preparation, W.J.; writing—review and editing, X.H., X.Y. and Z.F.; visualization, W.J.; supervision, X.H. All authors have read and agreed to the published version of the manuscript.

**Funding:** This research is financially supported by the National Key R&D plan Foundation of China (Grant No. 2021YFB2601300).

**Institutional Review Board Statement:** Not applicable.

**Informed Consent Statement:** Informed consent was obtained from all subjects involved in the study.

**Data Availability Statement:** The data that support the findings of this study are available from the corresponding author upon reasonable request.

**Conflicts of Interest:** The authors declare no conflict of interest.

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
