# Peer review of "Configuration Planning of Expressway Self-Consistent Energy System Based on Multi-Objective Chance-Constrained Programming"

_sustainability, doi:10.3390/su15065605_

Round 1
Reviewer 1 Report
Dear Authors,
Your paper is well written and presented.
Please see below some comments regarding your paper:
1) In table 5 please what are the meaning of the tyoe 1 wind turbine and type 2 wind turbine what are those values refering for (what metric???), the same for energy storage batteries....
2) Figure 1 please lable axis x what it is (k parameter)
3) I think that your conclusions should be sent to the discussion part of your paper; and a new conclusion should be writte describing what you advance to close the gap that you indicate in the introduction part with your method and suggestion for future works focusing in sustainability.
Author Response
First of all, thank you for taking time out of your busy schedule to review my article, your response was particularly professional and I accepted it willingly. Here are my answers and corrections to some of the points you raised:
Point 1: In table 5 please what are the meaning of the type 1 wind turbine and type 2 wind turbine what are those values refering for (what metric???), the same for energy storage batteries....
Response 1: The Type I wind turbine, Type II wind turbine, and energy storage battery in Table 5 refer to the number of configurations, and we have corrected them to be number of Type 1 wind turbine, number of Type 2 wind turbine and number of Energy storage batteries, respectively.
Point 2: Figure 10 please label axis x what it is (k parameter)
Response 2: Corrections completed.
Point 3: I think that your conclusions should be sent to the discussion part of your paper; and a new conclusion should be written describing what you advance to close the gap that you indicate in the introduction part with your method and suggestion for future works focusing in sustainability.
Response 3: We have rewritten the Conclusions section and changed the name of this section to Conclusions and future works. In this section, we reviewed and summarized the conclusions of this article, and added suggestions for future work at the end.

Reviewer 2 Report
Paper ID: sustainability-2302893
Title of the paper: Configuration planning of expressway self-consistent energy system based on multi-objective chance-constrained programming
Thank you for inviting me as a reviewer for the paper. I read this paper with interest. The manuscript is well organized and the contents fit with the journal’s topics. The methodology is well described and applied. The quality of the work clearly shows that the authors are very familiar with the topic of the paper.
In this paper, the authors studied the multi-objective capacity optimization problem of expressway self-consistent energy system based on the uncertainty of wind, solar radiation, and electrical load. In the presented model, the authors use several methods. The Pareto optimal solution set is obtained by non-dominated sorting genetic algorithm-II, and the optimal configuration scheme is given by CRITIC-TOPSIS optimization method.
My comments are as follows:
1) Add keywords: CRITIC, TOPSIS, NSGA-II (it is not obligatory, but I think that will be useful)
2) The abbreviation should be avoided from the abstract.
3) The authors should separate the literature analysis from the Introduction section and review more studies. Analyze new research. Add 5-15 recent papers (2021-2023), such as: Mukhametzyanov, I. (2021). The specific character of objective methods for determining weights of criteria in MCDM problems: Entropy, CRITIC, and SD. Decision Making: Applications in Management and Engineering, 4(2), 76-105; Arora, H., & Naithani, A. (2022). Significance of TOPSIS approach to MADM in computing exponential divergence measures for pythagorean fuzzy sets. Decision Making: Applications in Management and Engineering, 5(1), 246-263.
4) Show the advantages and limitations of the proposed methodology and this study in detail.
5) Expand the future direction by discussing related work.
Author Response
First of all, thank you for taking time out of your busy schedule to review my article, your response was particularly professional and I accepted it willingly. Here are my answers and corrections to some of the points you raised:
Point 1: Add keywords: CRITIC, TOPSIS, NSGA-II (it is not obligatory, but I think that will be useful)
Response 1: We have adjusted the keywords of the article.
Point 2: The abbreviation should be avoided from the abstract.
Response 2: Completed format corrections in abstract.
Point 3: The authors should separate the literature analysis from the Introduction section and review more studies. Analyze new research. Add 5-15 recent papers (2021-2023), such as: Mukhametzyanov, I. (2021). The specific character of objective methods for determining weights of criteria in MCDM problems: Entropy, CRITIC, and SD. Decision Making: Applications in Management and Engineering, 4(2), 76-105; Arora, H., & Naithani, A. (2022). Significance of TOPSIS approach to MADM in computing exponential divergence measures for pythagorean fuzzy sets. Decision Making: Applications in Management and Engineering, 5(1), 246-263.
Response 3: We have specifically divided the Introduction into three components: Background, Literature review, and Contributions and paper organization. In the literature review section and the Implementation of the proposed algorithm section in Section 4, we have added eight recent papers to illustrate that CRITIC-TOPSIS can be applied to the research content of this article.
Point 4: Show the advantages and limitations of the proposed methodology and this study in detail.
Response 4: We have rewritten the Conclusions section and changed the name of this section to Conclusions and future works. In this section, we review and summarize the conclusions of this article, and proposed the limitation that the case study in this article is relatively simple and the optimization results are highly dependent on the specific and detailed resource data of wind speed and solar radiation as well as electrical load demand of the planning area, which may be difficult to obtain.
Point 5: Expand the future direction by discussing related work.
Response 5: In Conclusions and future works section, we added suggestions for future work.
